# TOWARDS BETTER MULTI-HEAD ATTENTION VIA CHANNEL-WISE SAMPLE PERMUTATION

## ABSTRACT

Transformer plays a central role in many fundamental deep learning models, e.g., the ViT in computer vision and the BERT and GPT in natural language processing, whose effectiveness is mainly attributed to its multi-head attention (MHA) mechanism. In this study, we propose a simple and novel channel-wise sample permutation (CSP) operator, achieving a new structured MHA with fewer parameters and lower complexity. Given an input matrix, CSP circularly shifts the samples of different channels with various steps and then sorts grouped samples of each channel. This operator is equivalent to implicitly implementing cross-channel attention maps as permutation matrices, which achieves linear complexity and suppresses the risk of rank collapse when representing data. We replace the MHA of some representative models with CSP and test the CSP-based models in several discriminative tasks, including image classification and long sequence analysis. Experiments show that the CSP-based models achieve comparable or better performance with fewer parameters and lower computational costs than the classic Transformer and its state-of-the-art variants. The code is available at https://anonymous.4open.science/r/CSP-BA52.

## 1 INTRODUCTION

Transformer (Vaswani et al., 2017) has been widely adopted in the deep learning domain. Recent large language models like GPT (Brown et al., 2020; Radford et al.) and LLaMA (Touvron et al., 2023a;b) series are built based on the Transformer and its variants, which demonstrate their remarkable abilities in natural language processing. In the field of computer vision, Vision Transformers (ViTs) (Dosovitskiy et al., 2021), such as EfficientViT (Cai et al., 2023; Liu et al., 2023) and SHViT (Yun & Ro, 2024), exhibit exceptional performance and consistently push their limits. In addition, the Transformer-based models have been designed for the complex structured data in various applications, including the Informer (Zhou et al., 2021) for time series broadcasting, the Transformer Hawkes process (Zuo et al., 2020) for continuous-time event sequence prediction, the Graphormer (Ying et al., 2021) for molecular representation, the Mesh Transformer (Lin et al., 2021) for 3D mesh representation, the Set-Transformer (Lee et al., 2019) and Point-Transformer (Zhao et al., 2021) for point cloud modeling, and so on. Although some new alternatives like Mamba (Gu & Dao, 2023) and RWKV (Peng et al., 2023) have been proposed and shown their competitiveness in some aspects, Transformer still maintains a dominant position when developing deep learning models because of its strong performance and outstanding universality.

The effectiveness of Transformer is mainly attributed to its multi-head attention (MHA) mechanism (Vaswani et al., 2017). However, MHA's quadratic complexity concerning sequence length leads to a heavy, even unaffordable, computational overhead when modeling long sequences. To improve the efficiency of MHA, many variants of Transformer introduce sparse or low-rank structures into attention maps (Child et al., 2019; Kitaev et al., 2020; Wang et al., 2020; Ma et al., 2021; Wang et al., 2024) and apply algorithms friendly to GPU acceleration (Dao et al., 2022; Dao, 2024). At the same time, many attempts have been made to explore the mathematical reasons for the power of MHA, e.g., analyzing the representation power and rank collapse risk of MHA (Dong et al., 2021; Ying et al., 2021) and revisiting attention maps through the lens of kernel theory (Tsai et al., 2019; Qin et al., 2022) and optimal transport (Tay et al., 2020; Sander et al., 2022). Currently, the above two research directions seem "parallel" in most situations: The acceleration methods of MHA are

Figure 1: An illustration of the proposed channel-wise sample permutation operator and the equivalent implicit cross-channel attention maps.

often empirical, but the theoretical work mainly analyzes the classic MHA, making it seldom support the rationality of the accelerated MHAs or contribute to developing a new MHA.

In this study, we propose a novel **Channel-wise Sample Permutation (CSP)** operator, which leads to a new multi-head attention mechanism that is solid in theory and efficient in practice. As illustrated in Figure 1, given an input matrix, CSP first shifts the samples of different channels circularly with various steps and then sorts grouped samples of each channel. This operator is equivalent to implicitly implementing cross-channel attention maps as permutation matrices, which introduce inter- and intra-group interactions for the samples across different channels. CSP is much simpler than the classic MHA and its existing variants. It has no learnable parameters and can achieve linear computational complexity regarding sequence length.

The proposed CSP operator is motivated by the recent development of MHA. In particular, the work in (Child et al., 2019; Beltagy et al., 2020; Kitaev et al., 2020; Sander et al., 2022) empirically demonstrate the rationality of pursuing attention maps with sparse doubly stochastic structures, which is further verified by an analytic experiment in this study. CSP achieves permutation-based implicit attention maps that satisfy these structural properties, and thus, it has a good chance of providing a better MHA mechanism. Moreover, such attention maps have all-one spectrums because of their permutation nature. Based on the theoretical analysis framework provided in (Dong et al., 2021), we prove that replacing MHA with CSP can suppress the risk of rank collapse when representing data. In addition, we provide insightful understandings of the CSP operator by explaining its circular shifting and group sorting steps from the perspectives of optimal transport-based attention layer (Sander et al., 2022) and channel-wise mixer (Yu et al., 2022; Lian et al., 2022), respectively.

To demonstrate the usefulness of CSP, we replace the MHA of some state-of-the-art models with CSP and compare the CSP-based models with the original MHA-based ones in representative discriminative tasks, including long sequence analysis and image classification. For each model, replacing its MHA with CSP significantly reduces the number of parameters and the computational cost while maintaining or even improving model performance.

## 2 PRELIMINARIES AND RELATED WORK

Typically, given an input $\boldsymbol{X} \in \mathbb{R}^{N \times C}$, where $N$ indicates the length of a sequence or the size of a sample set and $C$ is the number of channels (feature dimensions), an attention head (Vaswani et al., 2017) first obtains the value, query, and key matrices by linear maps, i.e., $\boldsymbol{V} = \boldsymbol{X}\boldsymbol{W}_V \in \mathbb{R}^{N \times D}$, $\boldsymbol{Q} = \boldsymbol{X}\boldsymbol{W}_Q \in \mathbb{R}^{N \times D}$, and $\boldsymbol{K} = \boldsymbol{X}\boldsymbol{W}_K \in \mathbb{R}^{N \times D}$, and then projects $\boldsymbol{V}$ as follows:

$$\text{Att}(\boldsymbol{V}; \boldsymbol{Q}, \boldsymbol{K}) := \boldsymbol{P}(\boldsymbol{Q}, \boldsymbol{K})\boldsymbol{V}, \text{ where } \boldsymbol{P}(\boldsymbol{Q}, \boldsymbol{K}) = \text{Softmax}\Big(\frac{\boldsymbol{Q}\boldsymbol{K}^{\top}}{\sqrt{D}}\Big). \tag{1}$$

Here, we denote $\boldsymbol{V}$ as the input matrix of the head and $\boldsymbol{P}(\boldsymbol{Q}, \boldsymbol{K}) \in \mathbb{R}^{N \times N}$ as the attention map parametrized by $\boldsymbol{Q}$ and $\boldsymbol{K}$, respectively. The multi-head attention mechanism applies a group of linear maps, i.e., $\theta = \{\boldsymbol{W}_{V,m}, \boldsymbol{W}_{Q,m}, \boldsymbol{W}_{K,m} \in \mathbb{R}^{C \times D}\}_{m=1}^{M}$, to construct $M$ attention heads and concatenates their outputs, i.e.,

$$\text{MHA}_{\theta}(\boldsymbol{X}) := \|_{m=1}^{M}\text{Att}(\boldsymbol{V}_m; \boldsymbol{Q}_m, \boldsymbol{K}_m) \in \mathbb{R}^{N \times MD}, \tag{2}$$

where $\boldsymbol{V}_m = \boldsymbol{X}\boldsymbol{W}_{V,m}$, $\boldsymbol{Q}_m = \boldsymbol{X}\boldsymbol{W}_{Q,m}$, and $\boldsymbol{K}_m = \boldsymbol{X}\boldsymbol{W}_{K,m}$ for $m = 1, ..., M$, and "$\|$" denotes the concatenation operation. In practice, we set $MD = C$ for applying skip connections in the Transformer architecture, i.e., $\text{MHA}_\theta(\boldsymbol{X}) + \boldsymbol{X}$.

The attention map in (1) has quadratic computational complexity concerning the sequence length $N$ because of its "query-key-value" (abbreviately, QKV) architecture. Considering the high complexity per attention head, the MHA has to restrict the number of attention heads to achieve a trade-off between model capacity and computational efficiency, which may limit its representation power.

Many efforts have been made to improve the classic MHA. SparseTrans (Child et al., 2019) and Longformer (Beltagy et al., 2020) compute local attention maps based on the subsequences extracted by sliding windows, which leads to sparse global attention maps. To use shorter subsequences while retaining more information, S$^3$Attention (Wang et al., 2024) integrates global and local information by leveraging Fourier Transformation and a convolutional kernel. Some other models sparsify the key and query matrices directly by locality-sensitive hashing (LSH) (Kitaev et al., 2020) or ReLU (Qin et al., 2022). Besides pursuing sparse attention maps, Performer (Choromanski et al., 2021) and Linformer (Wang et al., 2020) apply low-rank attention maps. Recently, FlashAttention and its variants (Dao et al., 2022; Dao, 2024) further accelerate the computation of attention maps for long sequences by sophisticated I/O design, parallelism, and work partitioning. In addition to simplifying the computation of the attention maps, some work provides new understandings of the attention mechanism. The work in (Tsai et al., 2019; Choromanski et al., 2021; Qin et al., 2022) implements attention maps as various kernel matrices. The work in (Sander et al., 2022) implements doubly stochastic attention maps by the Sinkhorn-Knopp algorithm (Sinkhorn & Knopp, 1967) and explains the computation of each attention map as a discretized Wasserstein gradient flow.

Currently, the above accelerated or structured MHAs often lead to the performance degradation, while the theoretical understandings of MHA seldom help improve its computational efficiency in practice. Our work attempts to bridge the gap, proposing a theoretically solid multi-head attention mechanism with low complexity and competitive performance.

## 3 PROPOSED METHOD

### 3.1 MOTIVATION: PURSUING SPARSE DOUBLY STOCHASTIC ATTENTION MAPS

As shown in Section 2, many models apply various strategies to construct *sparse* attention maps, e.g., the locality-sensitive hashing (LSH) in (Kitaev et al., 2020), the subsequence sampling in (Child et al., 2019; Beltagy et al., 2020), and the sparse activation in (Qin et al., 2022). These models achieve encouraging performance and higher efficiency than the vanilla Transformer, demonstrating sparse attention maps' rationality. Besides making attention maps sparse, the work in (Sander et al., 2022) shows that in various discriminative tasks, the attention maps tend to be *doubly stochastic* automatically (i.e., $\boldsymbol{P} \in \Pi_N$, where $\Pi_N = \{\boldsymbol{A} \geq \boldsymbol{0} | \boldsymbol{A}\boldsymbol{1}_N = \boldsymbol{1}_N, \boldsymbol{A}^\top \boldsymbol{1}_N = \boldsymbol{1}_N\}$) during training,[1] and the Transformer applying doubly stochastic attention (called Sinkformer) outperforms the vanilla Transformer in image and text classification.

The above recent models show that sparse attention maps help improve the models' computational efficiency (thus making increasing attention heads feasible), and doubly stochastic attention maps help improve the models' discriminative power. These phenomena imply that **designing sparse doubly stochastic attention maps may lead to a better MHA mechanism and further boost model performance.** To verify this claim, we conduct an analytic experiment, replacing the attention maps in a standard ViT (Dosovitskiy et al., 2021) with simple permutation matrices (the doubly stochastic matrices with the strongest sparsity) and evaluating the model performance on the CIFAR-10 dataset (Krizhevsky, 2009). In particular, the ViT used in this experiment consists of six Transformer layers. Each Transformer has eight attention heads (i.e., $M = 8$), and each head sets $N = 64$, $C = 512$, and $D = 64$. For each layer, we replace the attention map of the $m$-th head with the following permutation matrix:

$$\boldsymbol{S}_{C(m-1)/D} = \begin{bmatrix} \boldsymbol{0} & \boldsymbol{I}_{C(m-1)/D} \\ \boldsymbol{I}_{N-C(m-1)/D} & \boldsymbol{0} \end{bmatrix}, \text{ for } m = 1, ..., M, \quad (3)$$

---

[1] Please refer to Section 3 in (Sander et al., 2022) for more details.

Table 1: A comparison for various MHAs and their classification accuracy (%) on CIFAR-10.

| MHA | #Heads per layer | Parameters per layer | Top-1 Acc. | Top-5 Acc. |
|---|---|---|---|---|
| $\|_{m=1}^{M} \boldsymbol{P}(\boldsymbol{Q}_m, \boldsymbol{K}_m)\boldsymbol{V}_m$ | $8\ (=M)$ | $\{\boldsymbol{W}_{Q,m}, \boldsymbol{W}_{K,m}, \boldsymbol{W}_{V,m}\}_{m=1}^{M}$ | 81.90 | 98.85 |
| $\|_{m=1}^{M} \boldsymbol{S}_{C(m-1)/D}\boldsymbol{V}_m$ | $8\ (=M)$ | $\{\boldsymbol{W}_{V,m}\}_{m=1}^{M}$ | 80.70 | 98.97 |
| $\|_{c=1}^{C} \boldsymbol{S}_{(c-1) \bmod N}\boldsymbol{v}_c$ | $64\ (=N)$ | $\{\boldsymbol{W}_{V,m}\}_{m=1}^{M}$ | **83.84** | **99.27** |

where $\boldsymbol{I}_N$ indicates an identity matrix with a size $N \times N$. Obviously, the permutation matrix $\boldsymbol{S}_{C(m-1)/D}$ corresponds to a circular shifting operator — $\boldsymbol{S}_{C(m-1)/D}\boldsymbol{V}_m$ means shifting the rows of $\boldsymbol{V}_m$ circularly with $C(m-1)/D$ steps. Furthermore, for each layer, we can concatenate $\{\boldsymbol{V}_m\}_{m=1}^{M}$ to get $\boldsymbol{V} = [\boldsymbol{v}_1, ..., \boldsymbol{v}_C] \in \mathbb{R}^{N \times C}$ and circularly shift the channels of this matrix by applying $\boldsymbol{S}_{(c-1) \bmod N}\boldsymbol{v}_c$ for $c = 1, ..., C$, where "$\mathrm{mod}$" is the modulo operation. In this case, the number of attention heads, equal to the number of distinguishable permutation matrices, becomes $N$. As shown in Table 1, even if the sparse doubly stochastic attention maps we designed are extremely simple and have no parameters, applying them with a sufficient number can still result in competitive, even better performance. This experimental result motivates us to construct sufficiently many sparse doubly stochastic attention maps with low complexity, leading to the proposed channel-wise sample permutation operator.

### 3.2 Channel-wise Sample Permutation for Implicit Cross-Channel Attention

As shown in Figure 1, given an input matrix $\boldsymbol{X} \in \mathbb{R}^{N \times C}$, the CSP operator first projects $\boldsymbol{X}$ to a value matrix with the same size, i.e., $\boldsymbol{V} = \boldsymbol{X}\boldsymbol{W} = [\boldsymbol{v}_1, ..., \boldsymbol{v}_C] \in \mathbb{R}^{N \times C}$, where $\boldsymbol{W} \in \mathbb{R}^{C \times C}$ and $\boldsymbol{v}_c$ denotes the $N$ samples in the $c$-th channel. Given $\boldsymbol{V}$, the CSP operator shifts the samples of different channels circularly with various steps and then sorts grouped samples of each channel, i.e.,

$$\mathrm{CSP}_{\boldsymbol{W}}(\boldsymbol{X}) := \|_{c=1}^{C} \mathrm{GSort}_K(\boldsymbol{S}_{J_c}\boldsymbol{v}_c) = \|_{c=1}^{C} \boldsymbol{P}_c\boldsymbol{v}_c, \quad \text{where } \mathrm{GSort}_K(\boldsymbol{v}) = \|_{k=1}^{K} \mathrm{Sort}(\boldsymbol{v}^{(k)}). \tag{4}$$

Here, $\boldsymbol{S}_{J_c}$ is the circular shifting operator defined in (3). $\mathrm{GSort}_K(\boldsymbol{v})$ denotes grouping the elements of a vector $\boldsymbol{v}$ into $K$ parts, i.e., $\boldsymbol{v} = [\boldsymbol{v}^{(1)}; ...; \boldsymbol{v}^{(K)}]$, and sorting each part accordingly. When implementing the CSP operator, we take the first channel $\boldsymbol{v}_1$ as the reference in this study: The circular shifting of each $\boldsymbol{v}_c$ is with respect to $\boldsymbol{v}_1$, and the group sorting permutes the elements of $(\boldsymbol{S}_{J_c}\boldsymbol{v}_c)^{(k)}$ according to the element-wise order of $\boldsymbol{v}_1^{(k)}$, for $c = 2, ..., C$ and $k = 1, ..., K$.

The CSP operator is equivalent to implicitly implementing sparse doubly stochastic attention maps as permutation matrices, which builds interactions for the samples across different channels. As shown in (4), we denote each attention map as $\boldsymbol{P}_c$. For $\boldsymbol{v}_1$, $\boldsymbol{P}_1 = \boldsymbol{I}_N$. For the remaining $\boldsymbol{v}_c$, $\boldsymbol{P}_c$ can be decomposed into the following two parts:

$$\boldsymbol{P}_c = \boldsymbol{T}_c\boldsymbol{S}_{J_c} = \mathrm{BlkDiag}(\{\boldsymbol{T}_c^{(k)}\}_{k=1}^{K})\boldsymbol{S}_{J_c}, \text{ for } c = 2, ..., C, \tag{5}$$

where $\boldsymbol{T}_c$ is a block-diagonal permutation matrix determined by the group sorting operation. The $k$-th block $\boldsymbol{T}_c^{(k)}$ is a permutation matrix determined by the sorting within the $k$-th group, which introduces intra-group sample interactions across different channels. The circular shifting operation introduces inter-group sampler interactions across different channels, and the ranges of the interactions are determined by the predefined shifting steps. As a result, for arbitrary two $\boldsymbol{v}_c$ and $\boldsymbol{v}_{c'}$, $\boldsymbol{P}_c\boldsymbol{v}_c$ and $\boldsymbol{P}_{c'}\boldsymbol{v}_{c'}$ captures their interactions determined by $\boldsymbol{P}_c^\top \boldsymbol{P}_{c'}$.

#### 3.2.1 Advantages over Existing MHAs

**High computational efficiency:** Replacing MHA with CSP leads to a new variant of Transformer. Table 2 compares the proposed model with the existing MHA-based models. We can find that the computational complexity of CSP can be $\mathcal{O}(N \log \frac{N}{K})$ when applying QuickSort (Hoare, 1962) to implement the group sorting operation, which is much lower than the computational complexity of the existing MHAs. When the group size is 2, we can achieve group sorting by the simple "$\mathrm{min}$-$\mathrm{max}$" operation (Anil et al., 2019; Tanielian & Biau, 2021), and the computational complexity further reduces to $\mathcal{O}(N)$. In addition, as shown in (4), except for the projection matrix $\boldsymbol{W}$ corresponding to the value matrix, CSP does not require additional projection matrices to construct the query and key matrices. In other words, its parameters are only one-third of the classic MHA.

Table 2: A comparison for existing MHA mechanisms and CSP.

| Model | Attention$(\boldsymbol{V}; \boldsymbol{Q}, \boldsymbol{K})$ | Complexity | Attention Structure |
|---|---|---|---|
| Transformer | Softmax$\left(\frac{\boldsymbol{Q}\boldsymbol{K}^\top}{\sqrt{D}}\right)\boldsymbol{V}$ | $\mathcal{O}(CN^2)$ | Row-normalized |
| SparseTrans | Local2D-Softmax$\left(\frac{\boldsymbol{Q}\boldsymbol{K}^\top}{\sqrt{D}}\right)\boldsymbol{V}$ | $\mathcal{O}(CN^{1.5})$ | Sparse+Row-normalized |
| Longformer | Local1D-Softmax$\left(\frac{\boldsymbol{Q}\boldsymbol{K}^\top}{\sqrt{D}}\right)\boldsymbol{V}$ | $\mathcal{O}(CNE)$ | Sparse+Row-normalized |
| Reformer | LSH-Softmax$\left(\frac{\boldsymbol{Q}\boldsymbol{K}^\top}{\sqrt{D}}\right)\boldsymbol{V}$ | $\mathcal{O}(CN\log N)$ | Sparse+Row-normalized |
| CosFormer | $(\boldsymbol{Q}_{\cos}\boldsymbol{K}_{\cos}^\top + \boldsymbol{Q}_{\sin}\boldsymbol{K}_{\sin}^\top)\boldsymbol{V}$ | $\mathcal{O}(\min\{CE_{QK}, NE_Q\})$ | Sparse |
| MEGA | $f\left(\frac{\boldsymbol{Q}\boldsymbol{K}^\top}{\sqrt{D}} + \boldsymbol{B}\right)\boldsymbol{V}$ | $\mathcal{O}(CN^2) \sim \mathcal{O}(CNr)$ | (Optional) Sparse+Row-normalized |
| Performer | $\phi_r(\boldsymbol{Q})\phi_r(\boldsymbol{K})^\top\boldsymbol{V}$ | $\mathcal{O}(CNr)$ | Low-rank |
| Linformer | Softmax$\left(\frac{\boldsymbol{Q}\psi_r(\boldsymbol{K})^\top}{\sqrt{D}}\right)\psi_r(\boldsymbol{V})$ | $\mathcal{O}(CNr)$ | Low-rank+Row-normalized |
| **Proposed** | $\|_{c=1}^C \boldsymbol{P}_c \boldsymbol{v}_c$ | $\mathcal{O}(CN\log\frac{N}{K}) \sim \mathcal{O}(CN)$ | Sparse+Doubly stochastic |

[1] "Local1D" considers subsequences with length $E$ when computing attention maps. "Local2D" considers the row-wise and column-wise local data for a sequence zigzagging in the 2D space.

[2] $\phi_r : \mathbb{R}^D \mapsto \mathbb{R}^r$, and $\phi_r(\boldsymbol{Q}), \phi_r(\boldsymbol{K}) \in \mathbb{R}^{N \times r}$; $\psi_r : \mathbb{R}^N \mapsto \mathbb{R}^r$, and $\psi_r(\boldsymbol{K}), \psi_r(\boldsymbol{V}) \in \mathbb{R}^{r \times D}$.

[3] $\boldsymbol{K}_{\cos} = \text{diag}(\{\cos\frac{\pi i}{2M}\}_{i=1}^N)\text{ReLU}(\boldsymbol{K})$, $\boldsymbol{K}_{\sin} = \text{diag}(\{\sin\frac{\pi i}{2M}\}_{i=1}^N)\text{ReLU}(\boldsymbol{K})$. So are $\boldsymbol{Q}_{\cos}$ and $\boldsymbol{Q}_{\sin}$. $E_{QK}$ and $E_Q$ are the numbers of nonzero elements in $\boldsymbol{Q}_{\cos}\boldsymbol{K}_{\cos}^\top$ and $\boldsymbol{Q}_{\cos}$, respectively.

[4] For MEGA, $\boldsymbol{B} \in \mathbb{R}^{N \times N}$ is a bias matrix. $f$ denotes the Softmax function in NLP tasks and a Laplace function in computer vision tasks. Its complexity becomes $\mathcal{O}(CNr)$ when applying a chunk mechanism to derive sparse attention maps.

**A low risk of rank collapse:** Besides significantly improving computational efficiency, CSP can suppress an ordinary risk of the classic MHA, rank collapse. In particular, we define the rank-1 estimation residual of a matrix $\boldsymbol{X}$ associated with an arbitrary matrix norm as

$$\epsilon(\boldsymbol{X}) = \boldsymbol{X} - \boldsymbol{1}\hat{\boldsymbol{x}}^\top, \text{ where } \hat{\boldsymbol{x}} = \arg\min_{\boldsymbol{x}} \|\boldsymbol{X} - \boldsymbol{1}\boldsymbol{x}^\top\|. \tag{6}$$

In addition, for a matrix $\boldsymbol{X} = [x_{nc}] \in \mathbb{R}^{N \times C}$, we can define its $(1, \infty)$-norm as $\|\boldsymbol{X}\|_{1,\infty} = \sqrt{\|\boldsymbol{X}\|_1 \|\boldsymbol{X}\|_\infty}$, where $\|\boldsymbol{X}\|_1 = \max_c \sum_{n=1}^N |x_{nc}|$ and $\|\boldsymbol{X}\|_\infty = \max_n \sum_{c=1}^C |x_{nc}|$, respectively. It has been known that $\|\epsilon(\boldsymbol{X})\|_{1,\infty}$ measures the rank collapse of $\boldsymbol{X}$ effectively, i.e., $\|\epsilon(\boldsymbol{X})\|_{1,\infty} \to 0$ means that $\boldsymbol{X}$ collapses to a rank-1 matrix. The work in (Dong et al., 2021) shows that if we construct a Transformer by stacking MHA layers without skip connections, its output matrix will lose its rank doubly exponentially with depth, i.e., $\|\epsilon(\text{MHA}_L \circ \cdots \circ \text{MHA}_1(\boldsymbol{X}))\|_{1,\infty} = \mathcal{O}(\|\epsilon(\boldsymbol{X})\|_{1,\infty}^{3^L})$, where $L$ is the number of the MHA layers.

Applying CSP can suppress this risk, which is supported by the following theorem.

**Theorem 1** *Suppose that we construct a layer-$L$ network as $(f \circ CSP)^L = (f_{\lambda_L} \circ CSP_{\boldsymbol{W}^{(L)}}) \circ \cdots \circ (f_{\lambda_1} \circ CSP_{\boldsymbol{W}^{(1)}})$. For $\ell = 1, ..., L$, $CSP_{\boldsymbol{W}^{(\ell)}}$ is a $C$-channel CSP operator, and $f_{\lambda_\ell} : \mathbb{R}^C \mapsto \mathbb{R}^C$ is a $\lambda_\ell$-Lipschitz function. Denote $\beta = \max_\ell \|\boldsymbol{W}^{(\ell)}\|_1$ and $\lambda = \max_\ell \lambda_\ell$. Then, we have*

$$\|\epsilon((f \circ CSP)^L(\boldsymbol{X}))\|_{1,\infty} \leq C^{\frac{L}{2}}(\lambda\beta)^L \|\epsilon(\boldsymbol{X})\|_{1,\infty}, \forall \boldsymbol{X} \in \mathbb{R}^{N \times C}. \tag{7}$$

Theorem 1 indicates a linear convergence rate of the residual. It means that the model applying CSP avoids the rapid decay of the matrix rank. A detailed proof is shown in Appendix A.

### 3.2.2 IMPLEMENTATION DETAILS

**Circular shifting:** The shifting step is crucial for the circular shifting operation, which determines the range of sample interaction. When the sequence length is comparable to the number of channels in each layer, i.e., $N \approx C$, we can simply set the shifting step size $J_c = c\lceil\frac{N}{C}\rceil$ for $c = 1, ..., C$, so that the circular shifting operation can generate sufficient distinguishable attention heads with respect to the sequence length. However, for long sequences, i.e., $N \gg C$, we need to set the shifting steps of different channels with high dynamics, making $C$ attention heads build diverse

interactions in a long sequence. In this study, given a Transformer-based model with $L$ layers, we consider all $L$ value matrices in these layers jointly, and set $LC$ different shifting steps based on power law, as illustrated in Figure 2. In particular, we denote $J$ as the base shifting step. For $c = 1, ..., LC$, we shift the $c$-th channel circularly with $J^{c-1} - 1$ steps. In addition, for the last channel, we require $J^{LC-1} - 1 \approx N - 1$. Therefore, we can set $J = \lfloor N^{1/(LC-1)} \rfloor$.

**Group sorting:** Instead of merely applying the circular shifting operation (as in Section 3.1), we introduce the group sorting operation to CSP, which helps increase the number of attention heads. Given an input matrix with size $N \times C$, the circular shifting operation constructs $\min\{N, C\}$ different attention maps, which results in repeated attention maps when $C > N$. For the channels applying the same circular shifting steps, the group sorting operation can make their attention maps different from each other as long as the orders of their samples are inconsistent. As a result, the group sorting helps CSP increase the number of attention heads from $\min\{C, N\}$ to $C$.

**A special case of CSP:** Note that when setting $K = 1$, the group sorting becomes the classic complete sorting, leading to a special case of CSP. Given $C$ channels, the complete sorting can directly generate at most $C$ distinguishable permutation matrices/attention heads. In addition, because of using the complete sorting, the circular shifting step of CSP becomes redundant. In the following experiments, empirically, implementing CSP as the complete sorting often works well when modeling long sequences while the CSP combining circular shifting with group sorting helps represent visual objects.

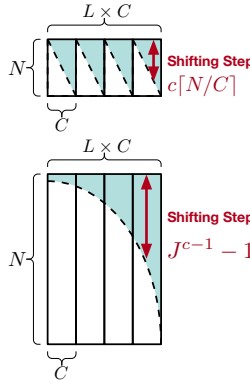

Figure 2: The shifting strategies when $N \approx C$ and $N \gg C$.

### 3.3 FUNCTIONALITY AND RATIONALITY ANALYSIS OF CSP

**Circular shifting works as a channel-wise mixer:** The circular shifting of CSP is similar to the channel-wise mixers used in visual representation models. In particular, the convolution neural networks like ShuffleNet (Zhang et al., 2018) and its variant (Ma et al., 2018) apply grouped convolution operation to reduce computational costs and increase inter-group interactions by shuffling the channels across different groups. This shuffling strategy inspires many lightweight channel-wise mixers, e.g., the hierarchical rearrangement in Hira-MLP (Guo et al., 2022), the spatial-shift module in S$^2$-MLP (Yu et al., 2022), and the axial-shift module in AS-MLP (Lian et al., 2022). For example, given a visual feature tensor with a size $H \times W \times C$ (i.e., 2D images with $C$ channels), the axial-shift module applies horizontal and vertical shifts with zero padding to the 2D images of different channels. The spatial-shift module first divides the input tensor into four parts by grouping its channels and then shifts the four sub-tensors along four different directions. Both these two modules apply small shifting steps to achieve local shifting. The circular shifting of CSP corresponds to applying these shifting modules to 1D sequences. To capture the short-range and long-range interactions between the samples of different channels simultaneously, we apply various shifting steps to different channels and replace zero padding with circular padding.

**Group sorting works as an optimal transport-based MHA:** Without causing any ambiguity, we denote $(\boldsymbol{S}_{J_c}\boldsymbol{v}_c)^{(k)}$ as $\boldsymbol{v}_c^{(k)}$ for simplification. It is easy to prove that the $\boldsymbol{T}_c^{(k)}$ in (5) is the optimal transport (OT) between $\boldsymbol{v}_1^{(k)}$ and $\boldsymbol{v}_c^{(k)}$, which can be derived by $\min_{\boldsymbol{T} \in \Pi_{N/K}} \langle -\boldsymbol{v}_1^{(k)}\boldsymbol{v}_c^{(k)\top}, \boldsymbol{T}\rangle$.[2] From this viewpoint, CSP achieves a new OT-based MHA mechanism. In addition, when approximating $\boldsymbol{T}_c^{(k)}$ as an entropic optimal transport, we can connect CSP to the doubly stochastic attention mechanism used in Sinkformer (Sander et al., 2022). In particular, each attention head in Sinkformer derives a doubly stochastic attention map, denoted as $\boldsymbol{T}_{T,\tau}$, by the Sinkhorn-Knopp algorithm (Sinkhorn & Knopp, 1967), i.e.,

$$\boldsymbol{T}_{t,\tau} = \text{Sinkhorn}_t\left(\exp\left(\frac{\boldsymbol{Q}\boldsymbol{K}^\top}{\tau\sqrt{D}}\right)\right), \text{ and } \boldsymbol{T}_{\infty,\tau} = \arg\min_{\boldsymbol{T} \in \Pi_N}\langle -\boldsymbol{Q}\boldsymbol{K}^\top, \boldsymbol{T}\rangle + \tau\sqrt{D}H(\boldsymbol{T}). \quad (8)$$

Here, $\text{Sinkhorn}_t(\boldsymbol{A})$ means normalizing the rows and columns of a nonnegative matrix $\boldsymbol{A}$ alternatively by $t$ times, i.e., $\boldsymbol{A}^{(0)} = \boldsymbol{A}$, and $\boldsymbol{A}^{(i)} = \text{N}_c \circ \text{N}_r(\boldsymbol{A}^{(i-1)})$ for $i = 1, ..., t$, where $\text{N}_c$ and $\text{N}_r$

---

[2]See Appendix B for a detailed derivation.

denote column-wise and row-wise normalization, respectively. As shown in (8), the attention map corresponds to the optimal solution of an entropic optimal transport problem (Cuturi, 2013) when $t \to \infty$, where $\langle \cdot, \cdot \rangle$ denotes the inner product operation, $H(\boldsymbol{T}) = \langle \boldsymbol{T}, \log \boldsymbol{T} \rangle$ denotes the entropy of $\boldsymbol{T}$, and its significance is controlled by $\tau > 0$.

We can connect Sinkformer to CSP by modifying the attention mechanism in (8) as follows. Given a value matrix $\boldsymbol{V} = [\boldsymbol{v}_1, ..., \boldsymbol{v}_C] \in \mathbb{R}^{N \times C}$, we replace the $\boldsymbol{Q}$ and $\boldsymbol{K}$ in (8) with $\boldsymbol{v}_1$ and $\boldsymbol{v}_c$, respectively, for $c = 1, ..., C$, and divide each vector into $K$ groups. We can achieve a $C$-head $K$-group doubly stochastic attention mechanism by applying the Sinkhorn-Knopp algorithm to $\boldsymbol{v}_1^{(k)} \boldsymbol{v}_c^{(k)\top}$, for $c = 1, ..., C$ and $k = 1, ..., K$, i.e.,

$$\boldsymbol{T}_{c,t,\tau} = \mathrm{BlkDiag}\left( \left\{ \mathrm{Sinkhorn}_t \left( \exp\left( \tfrac{1}{\tau} \boldsymbol{v}_1^{(k)} \boldsymbol{v}_c^{(k)\top} \right) \right) \right\}_{k=1}^K \right) = \mathrm{BlkDiag}(\{\boldsymbol{T}_{c,t,\tau}^{(k)}\}_{k=1}^K), \quad (9)$$

where $\boldsymbol{T}_{c,t,\tau}$ is a doubly stochastic attention map with a block-diagonal structure, and $\boldsymbol{T}_{c,t,\tau}^{(k)}$ denotes a local attention map, which corresponds to computing the entropic optimal transport between $\boldsymbol{v}_1^{(k)}$ and $\boldsymbol{v}_c^{(k)}$ when $t \to \infty$, i.e., $\boldsymbol{T}_{c,\infty,\tau}^{(k)} = \arg\min_{\boldsymbol{T} \in \Pi_{N/K}} \langle -\boldsymbol{v}_1^{(k)} \boldsymbol{v}_c^{(k)\top}, \boldsymbol{T} \rangle + \tau H(\boldsymbol{T})$.

The connection between the attention map in (9) and CSP is captured by the following theorem.

**Theorem 2** *If $\min_{\boldsymbol{T} \in \Pi_{N/K}} \langle -\boldsymbol{v}_1^{(k)} \boldsymbol{v}_c^{(k)\top}, \boldsymbol{T} \rangle$ admits a unique optimal solution, for $c = 1, ..., C$ and $k = 1, ..., K$, then for the $\boldsymbol{T}_{c,t,\tau}$ in (9), $\lim_{\tau \to 0} \boldsymbol{T}_{c,\infty,\tau} \boldsymbol{S}_{J_c}$ converges to the $\boldsymbol{P}_c$ in (5) weakly.*

Theorem 2 can be derived directly based on the weak convergence of entropic optimal transport (Theorem 5.10 in (Nutz, 2022)). This theorem indicates that a Sinkformer can implement CSP approximately if it $i$) sets the query and key matrices as the channels of the value matrix and $ii$) applies the Sinkhorn-Knopp algorithm to the grouped samples.

## 4 Experiments

To demonstrate the effectiveness and efficiency of CSP, we conduct comprehensive comparative and analytic experiments in two representative discriminative tasks, image classification and long sequence analysis. The implementation details can be found in Appendix C.

### 4.1 Image Classification

We conduct comparative experiments and ablation studies on three image datasets, including CIFAR-10, CIFAR-100 (Krizhevsky, 2009), and ImageNet-1k (Deng et al., 2009). For each dataset, we treat the classic ViT as the baseline and replace its MHA layers with $i$) **circular shifting**, $ii$) **group sorting**, and $iii$) the proposed **CSP** operator, respectively. Here, the circular shifting and the group sorting are two simplified CSP operators that help analyze the contributions of different CSP modules. Table 3 shows these models' size and classification accuracy. Applying CSP and its simplified variants can reduce the model size significantly without the query and key matrices. The circular shifting operator achieves competitive performance in all three datasets. In addition, although the standalone group sorting operator results in performance degradation, combining it with the circular shifting operator, i.e., the proposed CSP, can achieve the best performance. These observations are consistent with the experimental results achieved by mixer-MLP models (Yu et al., 2022; Lian et al., 2022): $i$) Simple channel-wise interactions can replace the dense and smoothed attention maps and lead to promising model performance, and $ii$) the shifting operator is crucial for CSP in computer vision tasks because it fully leverages the local similarity nature of the image. Moreover, when we increase the number of channels per layer and make the model size comparable to the original ViT, we can further boost the performance of the CSP-based models and achieve the best performance.

In Figure 3, we illustrate the singular spectrums of the output matrices achieved by different methods on ImageNet-1k. The spectrums achieved by the circular shifting and CSP operators decay much more slowly than the spectrum achieved by MHA. This observed phenomenon serves as a strong validation of the theoretical result in Theorem 1, providing further evidence that the representation model using permutation-based attention maps indeed carries a lower risk of rank collapse compared to the classic MHA-based model.

Table 3: The comparison for various models on the number of parameters ($\times 10^6$) and classification accuracy (%). The best result on each dataset is **bold**, and the second best result is underlined.

| Model | Attention | CIFAR-10 | | | CIFAR-100 | | | ImageNet-1k | | |
|-------|-----------|----------|-----|-----|-----------|-----|-----|-------------|-----|-----|
| | | #Param. | Top-1 | Top-5 | #Param. | Top-1 | Top-5 | #Param. | Top-1 | Top-5 |
| ViT | MHA | 9.52 | 81.90 | 98.85 | 9.65 | 53.30 | 79.97 | 22.05 | 76.53 | 92.81 |
| | Circular Shifting | 6.38 | 83.84 | 99.27 | 6.50 | 58.38 | 84.26 | 18.50 | 75.64 | 92.42 |
| | Group Sorting | 6.38 | 79.41 | 99.03 | 6.50 | 51.47 | 79.67 | 18.50 | 64.77 | 85.28 |
| **CSP (Proposed)** | | 6.38 | 84.81 | 99.35 | 6.50 | 59.16 | 84.76 | 18.50 | 76.66 | 93.05 |
| | | 9.52 | **85.02** | **99.37** | 9.65 | **59.23** | **85.09** | 22.05 | **77.14** | **93.23** |

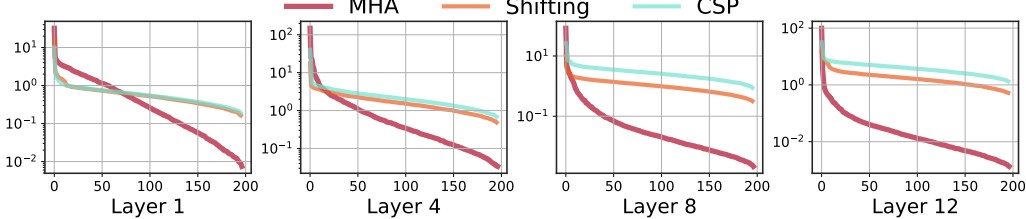

Figure 3: The singular spectrums of the output matrices achieved on ImageNet-1k.

## 4.2 LONG RANGE ARENA BENCHMARK

Long Range Arena (LRA) is a benchmark designed to evaluate models for long sequence analysis (Tay et al., 2021b), which consists of six discriminative tasks, including ListOps (Nangia & Bowman, 2018), byte-level text classification (Maas et al., 2011), byte-level document retrieval (Radev et al., 2013), and three sequentialized image classification tasks, i.e., CIFAR-10 (Krizhevsky, 2009), PathFind, and Path-X (Linsley et al., 2018).[3] Each image is formulated as a long sequence of pixels in the three image classification tasks. We first replace the MHA of the classic Transformer with CSP and compare it with other variants of Transformer. As shown in Figure 4 and the first part of Table 4, the Transformer using CSP outperforms other models on both performance and computational efficiency. It achieves the highest average score and the fastest training speed among all the models, and its memory cost is comparable to the most efficient variant of Transformer. For long sequence modeling, we simply implement CSP as the complete sorting operator in this experiment, which can capture the long-range interactions between the samples with the highest flexibility.

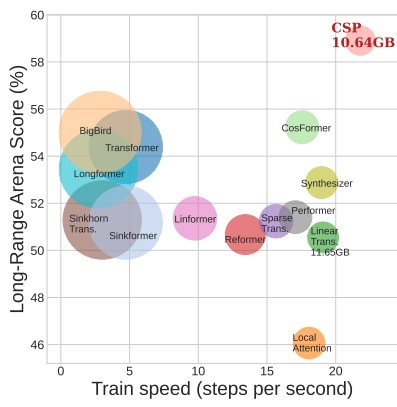

Figure 4: The performance and efficiency of various models on the LRA benchmark. The disk area indicates the memory cost of each method.

Besides improving the classic Transformer, we further plug CSP into the state-of-the-art attention-based model, MEGA (Ma et al., 2022), and analyze its impacts on the model performance. As evidenced in the second part of Table 4, the MEGA with dense attention maps currently outperforms all other methods, including those based on the state space model (SSM), such as S5 (Smith et al., 2022) and SPADE (Zuo et al., 2024), on the LRA benchmark. When MEGA applies chunked attention maps, its performance degrades slightly but its computational complexity can reduce from $\mathcal{O}(CN^2)$ to $\mathcal{O}(CNr)$, where $r$ is the chunk size. When replacing the attention mechanism of MEGA with the proposed CSP operator, the complexity of

---

[3]Given a set of gray-level images, each of which plots two points and several curves, PathFind aims to recognize whether there exists a path connecting the points in each image. Path-X is a more challenging version of Pathfind because of applying high-resolution images.

Table 4: Results (%) of various methods on the LRA benchmark. The first group contains the classic Transformer and its variants, and the second group contains the state-of-the-art methods on LRA. The best result on each dataset is **bold**, and the second best result is underlined.

| Type | Model | | ListOps | Text | Retrieval | Image | PathFind | Path-X | Avg. |
|---|---|---|---|---|---|---|---|---|---|
| MHA | Transformer (Vaswani et al., 2017) | | 36.37 | 64.27 | 57.46 | 42.44 | 71.40 | FAIL | 54.39 |
| | LocalAttention (Tay et al., 2021b) | | 15.82 | 52.98 | 53.39 | 41.46 | 66.63 | FAIL | 46.06 |
| | LinearTrans (Katharopoulos et al., 2020) | | 16.13 | **65.90** | 53.09 | 42.34 | 75.30 | FAIL | 50.55 |
| | Reformer (Kitaev et al., 2020) | | 37.27 | 56.10 | 53.40 | 38.07 | 68.50 | FAIL | 50.67 |
| | Sinkformer (Sander et al., 2022) | | 30.70 | 64.03 | 55.45 | 41.08 | 64.65 | FAIL | 51.18 |
| | SparseTrans (Child et al., 2019) | | 17.07 | 63.58 | 59.59 | 44.24 | 71.71 | FAIL | 51.24 |
| | SinkhornTrans (Tay et al., 2020) | | 33.67 | 61.20 | 53.83 | 41.23 | 67.45 | FAIL | 51.29 |
| | Linformer (Wang et al., 2020) | | 35.70 | 53.94 | 52.27 | 38.56 | 76.34 | FAIL | 51.36 |
| | Performer (Choromanski et al., 2021) | | 18.01 | 65.40 | 53.82 | 42.77 | 77.05 | FAIL | 51.41 |
| | Synthesizer (Tay et al., 2021a) | | 36.99 | 61.68 | 54.67 | 41.61 | 69.45 | FAIL | 52.88 |
| | Longformer (Beltagy et al., 2020) | | 35.63 | 62.85 | 56.89 | 42.22 | 69.71 | FAIL | 53.46 |
| | BigBird (Zaheer et al., 2020) | | 36.05 | 64.02 | 59.29 | 40.83 | 74.87 | FAIL | 55.01 |
| | Cosformer (Qin et al., 2022) | | **37.90** | 63.41 | 61.36 | 43.17 | 70.33 | FAIL | 55.23 |
| | **Transformer using CSP (Proposed)** | | 37.65 | 64.60 | **62.23** | **48.02** | **82.04** | FAIL | **58.91** |

| Type | Model | Complexity | ListOps | Text | Retrieval | Image | PathFind | Path-X | Avg. |
|---|---|---|---|---|---|---|---|---|---|
| CNN | CCNN (Romero et al., 2022) | $\mathcal{O}(CN^2)$ | 43.60 | 84.08 | FAIL | 88.90 | 91.51 | FAIL | 68.02 |
| SSM | ETSMLP (Chu & Lin, 2024) | | 62.55 | 88.49 | 86.72 | 75.34 | 91.66 | 93.78 | 83.09 |
| | S4 (Gu et al., 2022) | $\mathcal{O}(CN)$ | 58.35 | 76.02 | 87.09 | 87.26 | 86.05 | 88.10 | 80.48 |
| | S5 (Smith et al., 2022) | | 62.15 | 89.31 | **91.40** | 88.00 | 95.33 | **98.58** | 87.46 |
| | SPADE (Zuo et al., 2024) | | 59.70 | 87.55 | 90.13 | 89.11 | **96.42** | 94.22 | 86.19 |
| MHA | MEGA (Ma et al., 2022) | $\mathcal{O}(CN^2)$ | **63.14** | **90.43** | 91.25 | **90.44** | 96.01 | 97.98 | **88.21** |
| | MEGA-chunk (Ma et al., 2022) | $\mathcal{O}(CNr)$ | 58.76 | 90.19 | 90.97 | 85.80 | 94.41 | 93.81 | 85.66 |
| | **MEGA using CSP (Proposed)** | $\mathcal{O}(CN)$ | 61.85 | 90.27 | 90.09 | 87.42 | 93.74 | 91.98 | 85.89 |

the CSP-based MEGA becomes linear and thus comparable to that of the chunked MEGA and the SSM-based models. At the same time, the CSP-based MEGA is better than the chunked MEGA in the overall performance. These results serve as compelling evidence, demonstrating the practical rationality of CSP.

## 5 CONCLUSION & FUTURE WORK

We have proposed a novel channel-wise sample permutation operator, leading to a simple but effective surrogate of existing multi-head attention mechanisms. In theory, we demonstrate that the proposed CSP operator overcomes the rank collapse problem of the classic MHA because of implementing sparse doubly stochastic attention maps as permutation matrices. In addition, we explain the operator from the perspective of channel-wise mixer and optimal transport-based attention. For representative MHA-based models, replacing their MHA layers with the CSP operator helps improve their performance in discriminative tasks and reduce their computational cost at the same time. In summary, our work provides a promising solution to developing a better multi-head attention mechanism, demonstrating the usefulness of discrete algorithms like shifting and sorting in model design.

**Limitations and Future Work.** Currently, the design of CSP is motivated by pursuing sparse doubly stochastic attention maps, which restricts its application to discriminative tasks — the attention maps of Transformer decoder in generative tasks are lower-triangular, so that imposing the doubly stochastic constraint on the attention maps results in trivial identity matrices. For the Transformers in generative tasks (Radford et al.; Touvron et al., 2023a;b), how to achieve effective and efficient attention maps by simple algorithms is still an open problem, which is left as our future work. In addition, we implement our method based on Pytorch at the current stage. To maximize the computational efficiency of our method, we plan to refactor its underlying code and optimize its I/O, parallelism, and partitioning strategies as FlashAttention (Dao et al., 2022; Dao, 2024) did.

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

## A  THE PROOF OF THEOREM 1

**A Single Channel:** Given a CSP's output matrix, we can derive a residue for each channel as

$$\epsilon(\boldsymbol{P}_c \boldsymbol{X} \boldsymbol{w}_c) = \boldsymbol{P}_c \boldsymbol{X} \boldsymbol{w}_c - \mathbf{1}\hat{a}_c, \text{ for } c = 1, ..., C. \tag{10}$$

where $\hat{\boldsymbol{a}} = [\hat{a}_c]$ and

$$\hat{a}_c = \arg\min_a \|\boldsymbol{P}_c \boldsymbol{X} \boldsymbol{w}_c - \mathbf{1}a\| = \arg\min_a \|\boldsymbol{X} \boldsymbol{w}_c - \mathbf{1}a\|. \tag{11}$$

Then, we have

$$\|\epsilon(\boldsymbol{P}_c \boldsymbol{X} \boldsymbol{w}_c)\| = \|\boldsymbol{P}_c \boldsymbol{X} \boldsymbol{w}_c - \mathbf{1}\hat{a}_c\| = \|\boldsymbol{X} \boldsymbol{w}_c - \mathbf{1}\hat{a}_c\| \le \|(\boldsymbol{X} - \mathbf{1}\boldsymbol{x}^\top)\boldsymbol{w}_c\| \le \|\boldsymbol{w}_c\|\|\epsilon(\boldsymbol{X})\|, \tag{12}$$

where the second equation is based on the permutation invariance of the matrix norm, the first inequation is based on (11), and the second inequation is based on the sub-multiplicativity (or called consistency) of the matrix norm.

**A Single CSP:** Considering all $C$ heads and specifying the matrix norm to be 1-norm and $\infty$-norm, respectively, we have

$$\begin{aligned}
\|\epsilon(\text{CSP}_{\boldsymbol{W}}(\boldsymbol{X}))\|_1 &= \|(\|_{c=1}^C \boldsymbol{P}_c \boldsymbol{X} \boldsymbol{w}_c) - \mathbf{1}\hat{\boldsymbol{a}}^\top\|_1 \\
&= \max_c \|\boldsymbol{P}_c \boldsymbol{X} \boldsymbol{w}_c - \mathbf{1}\hat{a}_c\|_1 \\
&\le (\max_c \|\boldsymbol{w}_c\|_1)\|\epsilon(\boldsymbol{X})\|_1 \\
&= \|\boldsymbol{W}\|_1 \|\epsilon(\boldsymbol{X})\|_1.
\end{aligned} \tag{13}$$

$$\begin{aligned}
\|\epsilon(\text{CSP}_{\boldsymbol{W}}(\boldsymbol{X}))\|_\infty &= \|(\|_{c=1}^C \boldsymbol{P}_c \boldsymbol{X} \boldsymbol{w}_c) - \mathbf{1}\hat{\boldsymbol{a}}^\top\|_\infty \\
&\le \sum_{c=1}^C \|\boldsymbol{P}_c \boldsymbol{X} \boldsymbol{w}_c - \mathbf{1}\hat{a}_c\|_\infty \\
&\le \sum_{c=1}^C \|\boldsymbol{w}_c\|_\infty \|\epsilon(\boldsymbol{X})\|_\infty \\
&\le \sum_{c=1}^C \sum_{n=1}^N |w_{nc}| \|\epsilon(\boldsymbol{X})\|_\infty \\
&\le C(\max_c \|\boldsymbol{w}_c\|_1)\|\epsilon(\boldsymbol{X})\|_\infty \\
&= C\|\boldsymbol{W}\|_1 \|\epsilon(\boldsymbol{X})\|_\infty.
\end{aligned} \tag{14}$$

Combining the above two inequations, we have

$$\|\epsilon(\text{CSP}_{\boldsymbol{W}}(\boldsymbol{X}))\|_{1,\infty} \le \sqrt{C}\|\boldsymbol{W}\|_1 \|\epsilon(\boldsymbol{X})\|_{1,\infty}. \tag{15}$$

**A Single CSP followed by a Lipschitz function.** Given a Lipschitz function $f_\lambda : \mathbb{R}^C \mapsto \mathbb{R}^C$, we apply it to each row of a CSP's output matrix. For the residual of $f_\lambda \circ \text{CSP}_{\boldsymbol{W}}(\boldsymbol{X})$, we have

$$\begin{aligned}
\|\epsilon(f_\lambda \circ \text{CSP}_{\boldsymbol{W}}(\boldsymbol{X}))\| &= \|f_\lambda \circ \text{CSP}_{\boldsymbol{W}}(\boldsymbol{X}) - \mathbf{1}\hat{\boldsymbol{y}}^\top\| \\
&\le \|f_\lambda \circ \text{CSP}_{\boldsymbol{W}}(\boldsymbol{X}) - \mathbf{1}f_\lambda^\top(\hat{\boldsymbol{a}})\| \\
&\le \lambda \|\text{CSP}_{\boldsymbol{W}}(\boldsymbol{X}) - \mathbf{1}\hat{\boldsymbol{a}}^\top\| \\
&= \lambda \|\epsilon(\text{CSP}_{\boldsymbol{W}}(\boldsymbol{X}))\|,
\end{aligned} \tag{16}$$

where $\hat{\boldsymbol{y}} = \arg\min_{\boldsymbol{y}} \|f_\lambda \circ \text{CSP}_{\boldsymbol{W}}(\boldsymbol{X}) - \mathbf{1}\boldsymbol{y}^\top\|$ and $\hat{\boldsymbol{a}}$ is the vector associated to $\epsilon(\text{CSP}_{\boldsymbol{W}}(\boldsymbol{X}))$.

**Stacking $L$ CSP Operators:** We can recursively leverage the above results and derive the following inequation:

$$\begin{aligned}
\|\epsilon((f \circ \text{CSP})^L(\boldsymbol{X}))\|_{1,\infty} &\le \lambda_L \sqrt{C}\|\boldsymbol{W}^{(L)}\|_1 \|\epsilon((f \circ \text{CSP})^{L-1}(\boldsymbol{X}))\|_{1,\infty} \\
&\le C^{\frac{L}{2}}\Big(\prod_{\ell=1}^L \lambda_\ell \|\boldsymbol{W}^{(\ell)}\|_1\Big)\|\epsilon(\boldsymbol{X})\|_{1,\infty} \\
&\le C^{\frac{L}{2}}(\lambda\beta)^L \|\epsilon(\boldsymbol{X})\|_{1,\infty},
\end{aligned} \tag{17}$$

where $\beta = \max_\ell \|\boldsymbol{W}^{(\ell)}\|_1$ and $\lambda = \max_\ell \lambda_\ell$. When the $f$'s are the identity map, we have $\|\epsilon(\text{CSP}^L(\boldsymbol{X}))\|_{1,\infty} \le C^{\frac{L}{2}}\beta^L \|\epsilon(\boldsymbol{X})\|_{1,\infty}$.

# B  THE OT-BASED EXPLANATION OF CROSS-CHANNEL SORTING

For convenience, denote the group size $N/K$ by $G$. For $\boldsymbol{v}_1^{(k)}, \boldsymbol{v}_c^{(k)} \in \mathbb{R}^G$ The optimal transport distance between $\boldsymbol{v}_1^{(k)}$ and $\boldsymbol{v}_c^{(k)}$ can be defined as the following linear programming problem:

$$W(\boldsymbol{v}_1^{(k)}, \boldsymbol{v}_c^{(k)}) := \min_{\boldsymbol{T} \in \Pi_G} \langle \boldsymbol{D}, \boldsymbol{T} \rangle, \tag{18}$$

where $\boldsymbol{D} = (\boldsymbol{v}_1^{(k)} \odot \boldsymbol{v}_1^{(k)}) \mathbf{1}_G^\top + \mathbf{1}_G (\boldsymbol{v}_c^{(k)} \odot \boldsymbol{v}_c^{(k)})^\top - 2\boldsymbol{v}_1^{(k)} \boldsymbol{v}_c^{(k)\top}$ is the squared Euclidean distance matrix, and $\odot$ denotes the Hadamard product. Denote the optimal solution of (18) as $\boldsymbol{T}^*$. Because $\boldsymbol{T} \in \Pi_G$, we have

$$
\begin{aligned}
\boldsymbol{T}^* &= \arg\min_{\boldsymbol{T} \in \Pi_G} \langle \boldsymbol{D}, \boldsymbol{T} \rangle \\
&= \arg\min_{\boldsymbol{T} \in \Pi_G} \langle (\boldsymbol{v}_1^{(k)} \odot \boldsymbol{v}_1^{(k)}) \mathbf{1}_G^\top + \mathbf{1}_G (\boldsymbol{v}_c^{(k)} \odot \boldsymbol{v}_c^{(k)})^\top - 2\boldsymbol{v}_1^{(k)} \boldsymbol{v}_c^{(k)\top}, \boldsymbol{T} \rangle \\
&= \arg\min_{\boldsymbol{T} \in \Pi_G} \langle \boldsymbol{v}_1^{(k)} \odot \boldsymbol{v}_1^{(k)}, \boldsymbol{T}\mathbf{1}_G \rangle + \langle \boldsymbol{v}_c^{(k)} \odot \boldsymbol{v}_c^{(k)}, \boldsymbol{T}\mathbf{1}_G \rangle - 2\langle \boldsymbol{v}_1^{(k)} \boldsymbol{v}_c^{(k)\top}, \boldsymbol{T} \rangle \\
&= \arg\min_{\boldsymbol{T} \in \Pi_G} \underbrace{\langle \boldsymbol{v}_1^{(k)} \odot \boldsymbol{v}_1^{(k)}, \mathbf{1}_{G \times G} \rangle + \langle \boldsymbol{v}_c^{(k)} \odot \boldsymbol{v}_c^{(k)}, \mathbf{1}_{G \times G} \rangle}_{\text{A Constant } C_0} - 2\langle \boldsymbol{v}_1^{(k)} \boldsymbol{v}_c^{(k)\top}, \boldsymbol{T} \rangle \\
&\Leftrightarrow \arg\min_{\boldsymbol{T} \in \Pi_G} \langle -\boldsymbol{v}_1^{(k)} \boldsymbol{v}_c^{(k)\top}, \boldsymbol{T} \rangle.
\end{aligned}
\tag{19}
$$

In addition, because $\boldsymbol{v}_1^{(k)}$ and $\boldsymbol{v}_c^{(k)}$ are 1D vectors, their OT distance can be computed by aligning the elements of $\boldsymbol{v}_c^{(k)}$ to align to those of $\boldsymbol{v}_1^{(k)}$, which corresponds to the sorting operation, i.e.,

$$W(\boldsymbol{v}_1^{(k)}, \boldsymbol{v}_c^{(k)}) = \|\boldsymbol{v}_1^{(k)} - \boldsymbol{T}_c^{(k)} \boldsymbol{v}_c^{(k)}\|_2^2 = \underbrace{\langle \boldsymbol{v}_1^{(k)}, \boldsymbol{v}_1^{(k)} \rangle + \langle \boldsymbol{v}_c^{(k)}, \boldsymbol{v}_c^{(k)} \rangle}_{=C_0} - 2\langle \boldsymbol{v}_1^{(k)} \boldsymbol{v}_c^{(k)\top}, \boldsymbol{T}_c^{(k)} \rangle, \tag{20}$$

where $\boldsymbol{T}_c^{(k)}$ is the permutation matrix. Therefore, as long as $W(\boldsymbol{v}_1^{(k)}, \boldsymbol{v}_c^{(k)})$ has a unique optimal transport, $\boldsymbol{T}_c^{(k)} = \boldsymbol{T}^*$.

# C  IMPLEMENTATION DETAILS

## C.1  IMAGE CLASSIFICATION

The detailed hyperparameter setups are presented in Table 5. Both training and testing are conducted on 8 NVIDIA GeForce RTX 4080 SUPER GPUs.

Table 5: The hyperparameters of ViT using CSP on image classification tasks.

| Dataset | CIFAR-10 | CIFAR-100 | ImageNet-1k |
|---|---|---|---|
| #Groups $K$ | 32 | 128 | 98 |
| Shifting step | Linear | Linear | Linear |
| Batch Size | 64 | 64 | 256 |
| Epochs | 100 | 100 | 300 |
| Learning Rate | 1E-04 | 1E-04 | 5E-04 |
| LR scheduler | cosine | cosine | cosine |
| Optimizer | Adam | Adam | AdamW |
| Dropout Rate | 0.1 | 0.1 | 0.1 |
| Hidden Dims | 512 | 512 | 386 |
| Num. Layers | 6 | 6 | 12 |
| Pooling Type | mean | mean | mean |
| #Param. | 6.46M | 6.50M | 18.50M |

## C.2  LONG RANGE ARENA BENCHMARK

We strictly follow the LRA benchmark (Tay et al., 2021b)'s default data processing and experimental design. The detailed hyperparameter setups are presented in Table 6 and Table 7. For Image task,

Table 6: The hyperparameters of Transformer using CSP on LRA.

| Dataset | ListOps | Text | Retrieval | Image | PathFind |
|---|---|---|---|---|---|
| #Groups $K$ | 1 | 1 | 1 | 1 | 1 |
| Shifting step | — | — | — | — | — |
| Batch Size | 32 | 32 | 8 | 256 | 256 |
| Train steps | 5000 | 20000 | 5000 | 35156 | 125000 |
| Learning Rate | 5E-02 | 5E-02 | 5E-02 | 8E-03 | 1E-03 |
| LR scheduler | sqrt | sqrt | sqrt | cosine | cosine |
| Optimizer | Adam | Adam | Adam | Adam | Adam |
| Weight Decay | 1E-01 | 1E-01 | 1E-01 | 0 | 0 |
| Hidden Dims | 512 | 256 | 128 | 128 | 64 |
| Num. Layers | 4 | 4 | 4 | 4 | 6 |
| Pooling Type | cls | cls | cls | cls | cls |

Table 7: The hyperparameters of MEGA using CSP on LRA.

| Dataset | ListOps | Text | Retrieval | Image | PathFind | Path-X |
|---|---|---|---|---|---|---|
| #Groups $K$ | 1 | 1 | 1 | — | 512 | 8192 |
| Shifting step | — | — | — | Linear | Linear | Exp |
| Batch Size | 64 | 25 | 8 | 50 | 64 | 60 |
| Epochs | 60 | 50 | 40 | 200 | 200 | 100 |
| Learning Rate | 1E-03 | 4E-03 | 6E-03 | 1E-02 | 3E-02 | 1E-02 |
| LR scheduler | linear | linear | linear | linear | linear | linear |
| Optimizer | Adam | Adam | Adam | Adam | Adam | Adam |
| Weight Decay | 1E-02 | 1E-02 | 4E-02 | 2E-02 | 1E-02 | 1E-02 |
| Dropout Rate | 0.1 | 0.1 | 0.1 | 0.0 | 0.1 | 0.5 |
| Hidden Dims | 160 | 256 | 256 | 1024 | 256 | 128 |
| Num. Layers | 6 | 4 | 6 | 8 | 6 | 4 |
| Pooling Type | mean | mean | mean | mean | mean | mean |

we only apply the circular shifting operation. Both training and testing are conducted on 8 NVIDIA RTX A6000 GPUs.

In Figure 4, we compare Transformer using CSP with other baselines based on JAX Bradbury et al. (2018). These models are trained on 4 NVIDIA GeForce RTX 3090 GPUs. The detailed settings are as follows: The length of the sequence is 3K. The x-axis corresponds to the number of training steps per second. The y-axis corresponds to the average score (%) on the LRA benchmark. The peak memory usage of each model is represented as the area of the corresponding circle. For a better comparison, the values (GB) of the top-2 models are shown.

