# OpenReview forum: "Towards Better Multi-head Attention via Channel-wise Sample Permutation"
_ICLR.cc/2025/Conference — ICLR 2025 Conference Withdrawn Submission_

### Official Review · Reviewer_FDVQ · 2024-10-28

**Soundness:** 2
**Presentation:** 1
**Contribution:** 2
**Rating:** 3
**Confidence:** 3

**Summary:**

This paper presents a new module that aims to replace self-attention in Transformer layers. Specifically, this authors make an extreme use of channel shift operations to remove the quadratic computational overhead in Transformers as well as reducing the number of parameters. This method can be viewed as similar to MLP-mixer, which replaces the trainable channel mixing matrix of MLP-mixer to a permutation matrix that's driven according to their proposed algorithm. The algorithm has subquadratic theoretical complexity and achieves competitive results on a number of benchmarks.

**Strengths:**

Similar to ShuffleNet, this paper presents a new channel mixing method that shuffles channels of a given set of tokens then performs token-wise projection. This mechanism can remove the parameters necessary for explicit token mixing such as Attention, MLP-mixer, and SSM. The authors show that not only it leads to efficient number of parameters, but also have a mathematical strength over Attention that it is robust from a rank collapse. This paper demonstrates that this efficient approach performs on par, and even surpass previous methods  on some benchmarks.

**Weaknesses:**

- I mainly concern regarding the uncertainty of whether this method can be generally applicable. For example, as the authors mention in Sec 3.2.2, it is intractable to handle an input with a large number of tokens (e.g., long sequences) using this module, because of the limited number of channel dimension. The authors suggest a solution that incorporate multiple number of layers as a whole for channel shifting, but having two separate algorithms for different settings (N = C and N >> C) involves additional tuning of the model.

- Although I understand it could be due to lack of compute resources, the experiments are done in very light-weight configurations. Therefore, I am not sure if this method could be extended to a bit more general cases (e.g., 110M).

- I believe an ablation study of comparing to previous channel mixing methods is missing. For instance, I believe ShuffleNet is very similar to this paper in that it first shuffles channels then performs a token-wise projection. How much is this method better than the previous ones?

- Even though the main approach of this paper is very simple, it is confusing to understand due to the presentation of the paper. I believe it's mainly because of the ordering of the sections. Also, as Transformers indeed do have a residual connection, what are clear practical benefits of having a mixer that prevents from the rank collapse even if without a residual connection instead of Attention? Since the paper uses a lot of space for explaining mathematical analysis of the proposed module, I believe there should be a practical benefit of having such mathematical properties.

**Questions:**

- If this model is trained to sequences of length $L_{1}$, can it process sequences of $L_{2}$ where $L_{2} >> L_{1}$? Similarly, can this model be generalized to multiple $L$ settings, such as in the auto-regressive setting?
- Experimental details are not sufficient. When comparing to other methods, how are the experiments being controlled? Are every models using same number of layers? Are they parameter matched? For instance, how much is the number of parameters for each in table 4? In addition, I believe M2 [B] should be also added and SSM part should have S4D [C] and S4 with updated scores reported in [C].
- In Table 3, #Param of MHA is noted as 9.52 while CSP has two variants of 6.38 and 9.52. How are these #param matched? Specifically, the paper suggests that Q, K, V of MHA can be simplified to V only. Considering that given $D := d_model$, a transformer layer is $12D^2$, where $4D^2$ from MHA and $8D^2$ is from FFN, the proposed method should have $10D^2$ thanks to removing Q K projections in MHA. However, 6.38 is too small compared to 9.52, so I am not sure how these numbers are achieved.
- Why is Transformer so slow in Figure 4? Is it measured using FlashAttention?
- Does CSP also work on other tasks such as MQAR [B]?

[A] Fu et al. Monarch Mixer: A Simple Sub-Quadratic GEMM-Based Architecture\
[B] Arora et al. Zoology: Measuring and Improving Recall in Efficient Language Models\
[C] Gu et al. On the Parameterization and Initialization of Diagonal State Space Models

---

### Official Review · Reviewer_zwSe · 2024-10-28

**Soundness:** 3
**Presentation:** 3
**Contribution:** 2
**Rating:** 3
**Confidence:** 4

**Summary:**

This paper presents a new attention mechanism based on the multi-head self-attention. It introduces a channel-wise sample permutation operation, which circularly shifts the samples of different channels with various steps and then performs a sorting operation for the  grouped samples of each channel. The authors theoretically analyze the advantages of the proposed attention mechanism over the original self-attention. Experiments show that Transformers with the proposed attention mechanism performs well on several tasks, including image classification and long sequence analysis.

**Strengths:**

- The presentation of this paper is good. The authors clearly explain the motivation of this paper and give specific implementation details on the proposed method.

- The authors provide theoretical analysis to show the advantages of the proposed approach.

- Experimental results show that the proposed approach receives better results than previous attention mechanisms on CIFAR, ImageNet classification and long sequence analysis.

**Weaknesses:**

- It seems that the proposed approach performs better than other ViT variants as shown in Table 3. However, the authors did not compare the proposed approach with ViTs with other types of attention mechanisms, like the ones shown in Table 2.

- According to the appendix, the training recipe used in image classification is not new. It seems that the results in image classification is much less than vision transformers, like DeiT. Many popular training strategies are not used. Have the authors used stronger training recipes to conduct the image classification experiments?

- There are no analysis experiments to show how the proposed approach works. For example, how does the shifting step affect the model performance? Should the shifting step be set larger or smaller? There is no related experiments to demonstrate this.

- I do not think the experiments in this paper are comprehensive. The authors only show results on image classification and long sequence analysis. I suggest the authors to provide more results on more vision and language tasks to show the advantages of the proposed approach.

- The method text in Fig. 4 is too small. It is really difficult to identify the performance of different methods.

**Questions:**

The idea of this paper is good. However, the proposed approach is not experimentally verified. I suggest the authors to provide more experimental analysis to demonstrate the advantage of the proposed approach.

---

### Official Review · Reviewer_JJvA · 2024-11-03

**Soundness:** 3
**Presentation:** 2
**Contribution:** 2
**Rating:** 5
**Confidence:** 2

**Summary:**

In this paper, in order to improve the performance of Multi-head Attention, Channel-wise Sample Permutation is proposed , this method can reduce the number of Multi-head Attention parameters which reduces Multi-head Attention time complexity, and at the same time, it can reduce the risk of rank collapse and improve the stability of the training, the method is tested on several public datasets, and promising results are obtained.

**Strengths:**

1. Paper writing is clear and the methodology is more accurately described
2. The CSP module was used with good results, obtaining promising results in several tests.

**Weaknesses:**

1. The motivation of the modeling is clearer in the paper, and it can be understood that CSP needs to construct intergroup and intragroup permutations to achieve feature mixing, but there is some confusion in the paper's explanation of the model, and the linkage with the illustration (e.g., Fig. 1) is weak, which makes it more difficult to understand. For example, the specific definition of the substitution matrix $T_{c}^{k}$ and K in eq(5); also, unlike the definition in eq(3), $J_{c}$ seems to be independent of the number of heads, does it imply that shifting matrices are shared between different heads?
2. It seems that when $J_{c}$ is given, which parts of the whole model can be mixed between groups is determined; in other words, such a pre-determined mixing restricts the fusion between the features of the samples at any moment; moreover, unlike linear weighting, permutation makes each moment sample end up retaining only its own features or those of a certain other moment, and although the authors have demonstrated on some tests the algorithm's Even though the authors have demonstrated the effectiveness of the algorithm on some tests, the generalizability of the algorithm is questionable.
3. The authors have optimized the complexity of the model, but there are still many empirical settings in the model, such as the definition of shifting steps, the choice of K, and so on. These empirical modeling further introduces concerns about the utility of the model.

**Questions:**

see weakness

---

### Official Review · Reviewer_xgWD · 2024-11-03

**Soundness:** 3
**Presentation:** 2
**Contribution:** 3
**Rating:** 6
**Confidence:** 2

**Summary:**

This submission presents a novel channel-wise sample permutation operator (CSP operator) to achieve fewer parameters and lower complexity compared traditional multi-head attention mechanism. Specifically, CSP operator circularly shifts the samples of different channels with various steps and then sorts grouped samples of each channel. The experiments on several discriminative tasks demonstrate the proposed approach achieves comparable or better performance than the state-of-the-art Transformer variants.

**Strengths:**

1. The proposed operator achieves comparable performance to current Transformer variants with fewer parameters and lower complexity.
2. The experimental comparisons effectively demonstrate performance compared to current state-of-the-art Transformers.

**Weaknesses:**

1. The experiments are majorly evaluated on discriminative tasks.
2. According to Tab. 4, the improvement compared to previous approaches is limited.
3. The paper focus on the shift operator. I think it is necessary to discuss with previous shift operator, including those approaches in channel shift or spatial shift. e.g. TSM: Temporal Shift Module for Efficient and Scalable Video Understanding on Edge Device.

**Questions:**

I have a few questions about this paper.
1. In Tab. 3, do you use ViT-base as backbone for evaluation?
2. When the group size increases, will the sorting algorithm decline the speed in the real application in Cuda?

---

### Note · Authors · 2024-11-22

I have read and agree with the venue's withdrawal policy on behalf of myself and my co-authors.